# Quality of Life and Independent Factors Associated with Poor Digestive Function after Ivor Lewis Esophagectomy

**DOI:** 10.3390/cancers15235569

**Published:** 2023-11-24

**Authors:** Valerian Dirr, Diana Vetter, Thomas Sartoretti, Marcel André Schneider, Francesca Da Canal, Christian A. Gutschow

**Affiliations:** 1Department of Visceral and Transplant Surgery, University Hospital Zürich, 8032 Zürich, Switzerland; valerian.dirr@usz.ch (V.D.); diana.vetter@usz.ch (D.V.); marcelandre.schneider@usz.ch (M.A.S.); francesca.dacanal@spital-muri.ch (F.D.C.); 2Institute of Diagnostic and Interventional Radiology, University Hospital Zürich, 8032 Zürich, Switzerland

**Keywords:** esophageal cancer, gastric cancer, oncology, surgery, clinical trials, quality of life, digestive function, esophagectomy

## Abstract

**Simple Summary:**

Surgery is the backbone of curative treatment strategies, but it is associated with a high risk of reduced alimentary function and impaired health-related quality of life (HRQL). This study addresses the impact of Ivor Lewis esophagectomy on patients’ digestive function and seeks to identify independent factors associated with poor functional outcomes by assessing multiple dimensions of digestive performance (dysphagia, gastroesophageal reflux disease, delayed gastric conduit emptying, and dumping syndrome) and HRQL. Our research provides clinicians, patients, and surgeons with new insights into treatment approaches and postoperative care that may impact the management of esophageal cancer.

**Abstract:**

Transthoracic esophagectomy results in a radical change in foregut anatomy with multiple consequences for digestive physiology. The aim of this study was to identify factors associated with poor functional outcomes by assessing multiple dimensions of digestive performance and health-related quality of life (HRQL). Patients with cancer-free survival after Ivor Lewis esophagectomy were included. Four functional syndromes (dysphagia, gastroesophageal reflux disease (GERD), delayed gastric conduit emptying (DGCE), and dumping syndrome (DS)) and HRQL were assessed using specifically designed questionnaires. Patient outcomes were compared with healthy controls. Independent factors associated with poor digestive performance were identified through multivariable analysis. Sixty-five postoperative patients and 50 healthy volunteers participated in this study. Compared with controls, patients had worse outcomes for dysphagia, GERD, DS, and HRQL, but not for DGCE. A multivariate analysis showed a significant correlation of reduced digestive performance with ASA score, squamous cell carcinoma, open or hybrid surgical approach, and (neo)adjuvant therapy. In contrast, no individual patient factor was found to be associated with dumping syndrome. Digestive function and HRQL are substantially impaired after Ivor Lewis esophagectomy for cancer. Comorbid patients undergoing multimodal treatment and open access surgery for squamous cell carcinoma have the highest risk of poor functional outcome.

## 1. Introduction

Therapeutic concepts for esophageal cancer have evolved considerably over the past decades [1]. With the recent introduction of adjuvant immunotherapy, five-year survival rates of 60% and higher can be achieved even in advanced situations [2]. Surgery remains the backbone of curative treatment strategies but carries a high risk of poor digestive function and reduced health-related quality of life (HRQL) [3,4].

Poor functional performance after esophagectomy is usually multifactorial, partially owed to the changed foregut anatomy and directly linked to the surgical reconstruction technique, but also to other factors such as psycho-oncologic stress, comorbidities, functional deterioration, and physiologic aging [5,6,7,8]. Importantly, the surgical reconstruction itself may be subject to degeneration, including structural and functional changes in terms of conduit redundancy and dilation, conduit- and para-conduit herniation, or increasing acid production of the stomach as an esophageal substitute [9,10,11,12]. 

As survival after esophagectomy improves, symptom-based functional follow-up becomes increasingly important. In a previous paper, we identified four high-prevalence functional syndromes that are relevant for digestive function after esophagectomy: dysphagia, gastroesophageal reflux (GERD), delayed gastric conduit emptying (DGCE), and accelerated conduit emptying; the latter typically manifesting as dumping syndrome (DS) [8]. These functional syndromes each exhibit a specific set of symptoms, which may overlap considerably (Figure 1). Therapeutic options range from dietary counseling and symptom-based medication to various forms of endoscopic intervention and surgical revision [8]. Since published research has mostly focused exclusively on general postoperative HRQL without addressing organ-specific outcomes [13,14,15], the aim of this project was (a) to perform a comprehensive assessment of functional syndromes in mid- and long-term survivors after esophagectomy and (b) to identify factors associated with impaired outcomes. To increase stringency, we included only patients who underwent a standardized surgical approach with two-field lymphadenectomy, gastric tube reconstruction, and intrathoracic esophago-gastrostomy. This operation was introduced by the British surgeon Ivor Lewis in 1946 [16] and represents the current surgical standard for Western patients with esophageal cancer in the middle and lower thirds.

## 2. Materials and Methods

### 2.1. Patients, Own References, and EORTC Reference Cohort

Patients who underwent an Ivor Lewis esophagectomy between 2015 and 2020 and who had a cancer-free survival of ≥12 months were identified in a prospective institutional database. All patients were under active oncologic surveillance with biannual (≤3 years after esophagectomy) or annual (>3 years after esophagectomy) computed tomography, clinical visits, and endoscopy on demand. The study was approved by the Ethical Committee of the Canton of Zurich (BASEC Nr.: 2021-00329).

Own reference values for Eckardt-score, GERD-HRQL, DGCE-score, Sigstad-score, and EORTC OES-18- and QLQ-C30 scores were generated from a cohort of fifty healthy volunteers. All questionnaires were independently completed via an online tool (www.surveymonkey.com, accessed 22 December 2020). A published EORTC cohort of 7802 healthy volunteers served as an additional reference for the EORTC QLQ-C30 score [17].

### 2.2. Assessment of Functional Syndromes and HRQL

Functional syndromes were assessed using validated scoring systems: the Eckardt-score for dysphagia [18], the GERD-HRQL questionnaire [19], the DGCE-score [20], and the Sigstad-score for DS [21]. The Eckardt-score was calculated using the attribution of points to symptoms, ranging from 0 (no symptoms) to 12 (distinct symptoms). GERD-HRQL was calculated by summing the individual scores to each question, ranging from 0 (no symptom) to 5 (symptoms incapacitating daily activities), with a maximum possible score of 75. DGCE was calculated by the total symptom score ranging between 0 points (no symptoms) and 15 points (very much). Sigstad’s score was calculated by allocating points to symptoms. The total points were summarized into a calculated diagnostic index. A score above 7 was suggestive for DS.

The EORTC core questionnaire (QLQ C-30) and the esophageal cancer-specific module (QLQ OES-18) were used to collect HRQL data. Validation of the questionnaires has been previously described [22,23]. The QLQ C-30 version 3.0 includes nine multi-item scales: a Global Health Status (GHS) scale, five functional scales (physical, role, cognitive, emotional, and social functioning), three symptom scales (fatigue, pain, and nausea) and six single-item scales (dyspnea, insomnia, appetite loss, constipation, diarrhea, and financial difficulties). The QLQ OES-18 module’s questions transform into four symptom scales (eating, reflux, esophageal pain, and dysphagia) and six single items (cough, dry mouth, taste, choking, talking, and trouble swallowing saliva). Patients and healthy volunteers were contacted by phone and requested to answer the questionnaires via an email link or in print. Questionnaires were available in German and Italian languages. Missing data were completed through telephone interview or handled according to the EORTC scoring manual. Results of the EORTC questionnaires were scored by averaging the scale’s contributing items (raw score) and applying a linear transformation to normalize the range of the score from 0 to 100 [24]. Mean scores with standard deviations (SD) were calculated. For functional scores and global HRQL, higher scores represent better function and HRQL. In symptom scales, higher scores indicate worse symptoms.

### 2.3. Statistical Analysis

Results from the study population and the reference cohorts were compared using two-sided t-tests. In addition, multivariable analysis was performed using individual-patient variables to assess their impact on scoring results (i.e., dependent variable). The influence on the following scores were analyzed: QLQ C-30, QLQ OES-18, and functional syndromes (Eckardt-score, DGCE-score, GERD-HRQL score, Sigstad-score). Specifically, generalized linear models (GLM) using iteratively reweighted least squares to find the maximum likelihood estimates were fitted. GLMs were iteratively optimized based on the Akaike information criterion (AIC). The following variables (i.e., predictors) were included in the model: ASA, WHO/ECOG, ICU readmission (yes or no), readmission within 90 days post-op (yes or no), histology, neoadjuvant radio-/chemotherapy (no, only chemotherapy, chemo- and radiotherapy), adjuvant radio-/chemotherapy (no, only chemotherapy, only radiotherapy, chemo- and radiotherapy), UICC, surgical access (total MIC, hybrid, open), highest level of postoperative complications, number of postoperative complications, CCI at discharge, and follow-up shorter or longer than 36 months. Unless specified otherwise, two-sided *p*-values < 0.05 were considered significant. All statistical analyses were performed using R programming language.

## 3. Results

### 3.1. Basic Characteristics of Study Patients and Reference Cohorts

A total of 80 patients were eligible, and 65 (81% response rate) with a median (IQR) follow up of 29 (18–49) months after esophagectomy participated in this trial. At the time of the interview, the median (IQR) age was 71 (64–77) years, where 83% were male, and the median (IQR) BMI was 25.7 (24.2–28.5) kg/m^2^. Most patients had undergone minimal invasive esophagectomy (74%) for advanced stage (UICC III 46%) adenocarcinoma (65%) after neoadjuvant chemoradiation (56%). The healthy reference cohort had a median (IQR) age of 65 (58–71), where 54% were female and 46% male, and the median (IQR) BMI was 24.4 (22.6–28.4) kg/m^2^. The EORTC cohort included 7802 healthy volunteers (age 40–80 years; 52% men and 48% women). Clinical details of the study cohort and the control groups are summarized in Table 1.

### 3.2. Functional Syndromes: Dysphagia, GERD, DGCE, and DS

In univariate analysis, both Eckardt- and GERD-HRQL had significantly higher scores (indicating worse outcomes) in the study cohort compared with the healthy reference group. However, no statistically significant difference was found for the DGCE and DS scores (Figure 2).

In the multivariate analysis, the following variables were predictive of worse outcome: dysphagia correlated with a high ASA score (*p* = 0.02). Predictors for GERD were an open surgical approach (*p* = 0.01) and neoadjuvant therapy (*p* = 0.03). DGCE was associated with an open surgical approach (*p* = 0.04). In contrast, none of the clinical variables studied was predictive of DS occurrence (Table 2).

### 3.3. General (EORTC QLQ C-30) and Esophagus-Specific (EORTC OES-18) HRQL

In the univariate analysis, overall HRQL, as measured with the EORTC QLQ C-30, was generally lower in patients after esophagectomy compared with own and EORTC reference data (Figure 3). This difference was significant for the Global Health Score and physical, role, and social functions but did not reach statistical significance for emotional and cognitive functions. Likewise, esophagus-specific symptom scores of the EORTC OES-18 revealed significantly inferior results in esophagectomy patients compared with own reference (Figure 4). This difference was significant for all symptoms except choking and dry mouth.

In multivariate analysis, the following variables were predictive of worse outcome in general HRQL (QLQ C-30): high ASA score (*p* = 0.04), ICU readmission (*p* < 0.01), SCC (*p* < 0.01), adjuvant chemotherapy (*p* < 0.01), and hybrid surgical access (*p* < 0.01). Similarly, the following variables were predictive of lower esophagus-specific HRQL as measured with QLQ OES-18: SCC (*p* = 0.02), neoadjuvant chemotherapy (*p* = 0.02), hybrid (*p* = 0.02), and open surgical access (*p* < 0.01) (Table 2).

## 4. Discussion

Modern surgical therapy for esophageal cancer aims to cure patients with as little impact on quality of life as possible. However, this goal is difficult to achieve owing to the oncologic aggressiveness of the disease and the inherent surgical mutilation with significant changes in foregut anatomy and physiology. Poor postoperative digestive function can be attributed to a quartet of four common and often clinically overlapping functional syndromes, namely, dysphagia, GERD, DGCE, and DS [8].

In this regard, a particular strength of this study is the comprehensive approach to functional follow-up after esophagectomy: we monitored not only general HRQL but also the relevant digestive syndromes using validated questionnaires. Furthermore, we compared the results obtained in patients with those of two separate reference cohorts, and we identified independent factors associated with poor digestive function through a multivariable analysis. In addition, our study population was highly homogeneous, and the surgical procedure was well standardized. Moreover, all operations were performed by the same surgical team with stringent and high follow-up rates compared with previous research in the field [4,6,25]. Consequently, we believe that our study provides important and reliable new data on the relationship between comorbidities, surgical technical details, and functional outcome at follow-up.

Our study is in good agreement with previous publications confirming reduced overall and cancer-specific HRQL after esophagectomy [4,5,6,25,26]. Although improvement may occur over time [14,27,28,29] with approximately 50% of mid- and long-term survivors achieving similar HRQL compared with healthy reference subjects [4], it has been shown that reduced HRQL may persist long-term after esophagectomy [30,31,32,33,34], which is supported by our results beyond 12 months of follow up.

The current study confirms the relevance of “functional syndromes” after esophagectomy, with dysphagia, GERD-HRQL, and DS scores being substantially higher in patients compared with healthy controls, which is consistent with other research findings [33,34,35,36,37]. Interestingly, we did not find higher DGCE scores in patients compared with controls, indicating that the DGCE-score [21], which was specifically designed for patients after esophagectomy, may not be an ideal measure in healthy subjects.

Our study is unique in that we identified several clinical factors associated with poor functional performance after esophagectomy. Thus, open or hybrid surgical access, high ASA and ECOG scores, (neo)adjuvant treatment, ICU readmission, and SCC predicted impaired functional outcome. Hence, our results confirm the findings of others that open access surgery leads to worse functional outcomes [13,38]. Because long-term oncologic outcomes after open and minimally invasive procedures are equivalent [39], our results underscore the importance of less invasive surgical procedures in patients undergoing esophagectomy. Likewise, our results confirm other research findings [40,41] that comorbidities and poor performance status are important predictors of long-term HRQL impairment.

Our study has several limitations. First, we cannot exclude selection bias due to the retrospective design of our trial. However, 81% of patients responded with a complete set of data, leaving less than 20% of long-term survivors disregarded. In addition, there is an inevitable bias due to the considerable tumor-related death rate among esophageal cancer patients. Another relevant problem may be the fact that owing to the cross-sectional design of the study, we were not able to compare our data to pretreatment baseline HRQL. Finally, this study did not aim to collect direct physiologic or biochemical evidence of specific functional syndromes after esophagectomy, such as GERD-HRQL, DGCE, dumping syndrome, and dysphagia. Therefore, the next step in future research, ideally prospective, would be to focus on establishing a direct link between patient-reported symptoms and objective signs of impaired digestive function.

## 5. Conclusions

In conclusion, this study provides new insights into the functional consequences of esophageal cancer surgery. It highlights individual patient factors such as surgical approach, comorbidities, and squamous cell carcinoma that contribute to impaired HRQL, psychosocial well-being, and physiologic function during long-term follow up. We believe that our findings will benefit both clinicians and patients by facilitating therapeutic decisions and improving treatment pathways. In addition, our data are relevant for surgeons to understand which individuals are at risk for poor digestive function and to counsel their patients before cancer surgery.

## Figures and Tables

**Figure 1 cancers-15-05569-f001:**
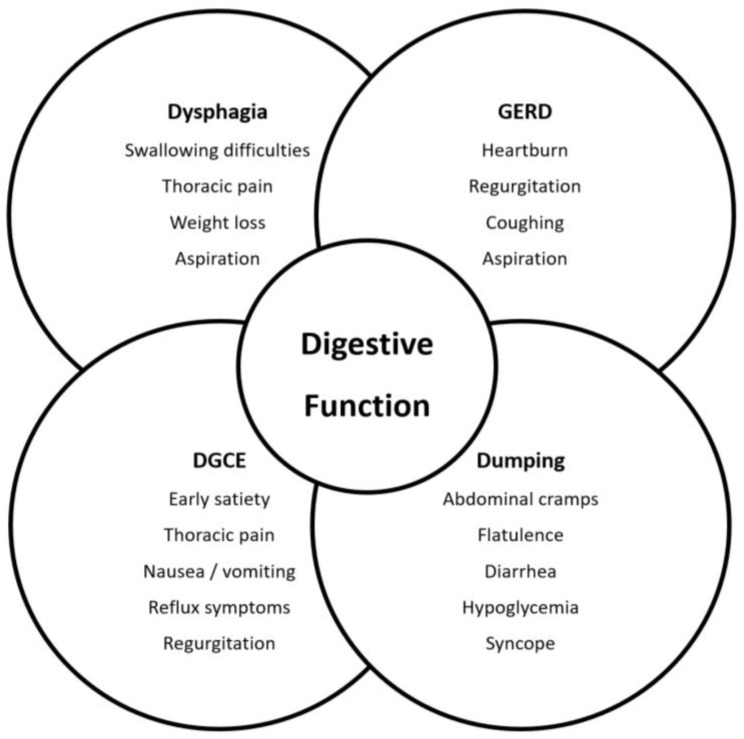
Venn diagram depicting functional syndromes after esophagectomy with typical symptom spectra and their relationship with digestive performance.

**Figure 2 cancers-15-05569-f002:**
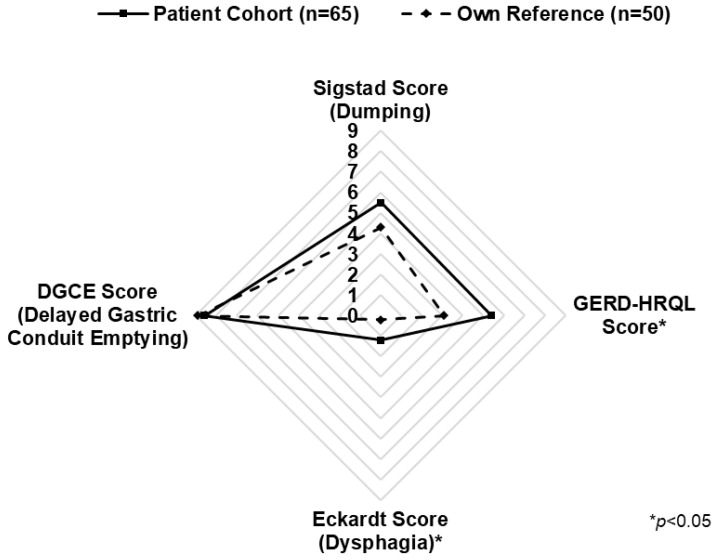
Spider graph representing mean scores for functional syndromes in patients after Ivor Lewis esophagectomy (solid line) and own reference (dashed line).

**Figure 3 cancers-15-05569-f003:**
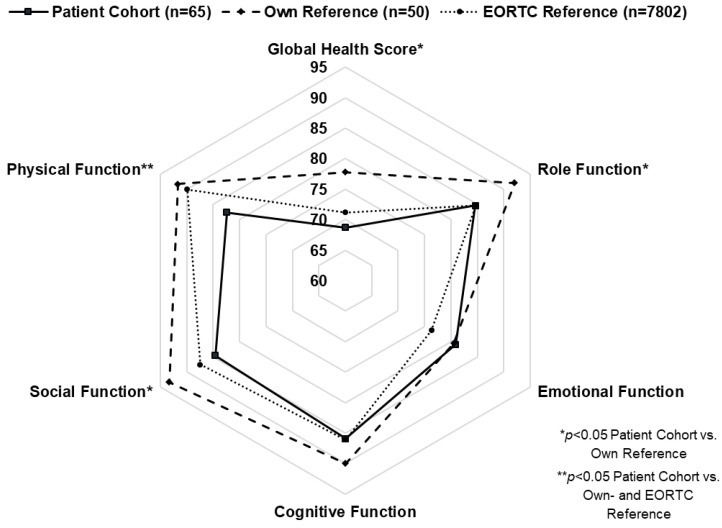
Spider graph representing mean scores for global health (GHS) and functional scales of the EORTC QLQ C-30 in patients after Ivor Lewis esophagectomy (solid line), own reference (dashed line), and in the EORTC reference (dotted line).

**Figure 4 cancers-15-05569-f004:**
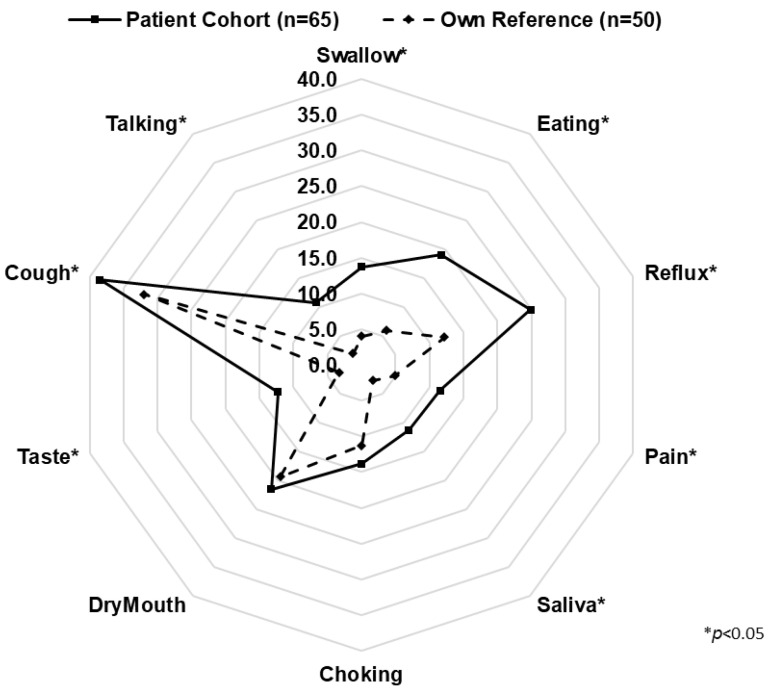
Spider graph representing mean scores for the EORTC QLQ OES-18 in patients after Ivor Lewis esophagectomy (solid line) and own reference (dashed line).

**Table 1 cancers-15-05569-t001:** Baseline characteristics of the patient cohort, own-, and EORTC reference.

Characteristic	Patient Cohort (n = 65)	Own Reference (n = 50)	EORTC Reference (n = 7802)
Age in years median (IQR)	71 ^1^ (64–77)	65 (58–71)	40–80 years
*Sex*			
Male	54 (83)	23 (46)	4057 (52)
Female	21 (17)	27 (54)	3745 (48)
BMI median (IQR)	25.7 (24.2–28.5)	24.4 (22.6–28.4)	
*Performance status*			
ASA 1 and ECOG 0	1 (2)	50 (100)	
ASA ≥ 2 and/or ECOG ≥ 1	64 (98)		
*Tumor histology*			
Adenocarcinoma	42 (65)		
Squamous cell carcinoma	16 (25)		
Other	7 (10)		
*Pathological tumor stage (n = 62)*			
UICC 0	8 (13)		
UICC I	12 (19)		
UICC II	10 (15)		
UICC III	28 (46)		
UICC IV	4 (7)		
*Neoadjuvant treatment*			
None	21 (32)		
Chemotherapy	8 (12)		
Chemoradiotherapy	36 (56)		
*Adjuvant treatment*			
None	51 (78)		
Chemotherapy	12 (18)		
Chemoradiotherapy	1 (2)		
Radiotherapy	1 (2)		
*Follow-up time months*			
Median (IQR)	29 (18–49)		
<36 months	35 (54)		
≥36 months	30 (46)		
*Surgical access*			
Total MIS	48 (74)		
Hybrid	13 (20)		
Open	4 (6)		
*90-day readmission*			
Yes	5 (8)		
No	60 (92)		
CCI at discharge median (IQR)	20.9 (0–29.6)		

^1^ Data are presented as n (%) unless otherwise indicated. EORTC, European Organization for Research and Treatment; ASA, American Society of Anesthesiologists; ECOG, Eastern Cooperative Oncology Group; UICC, Union for International Cancer Control; MIS, Minimal Invasive Surgery; CCI, Comprehensive Complication Index.

**Table 2 cancers-15-05569-t002:** Variables predicting worse overall (EORTC QLQ C-30), cancer-specific (EORTC QLQ OES-18), reflux-related HRQL, Eckardt-, DGCE-, and Sigstad score in multivariate analysis.

Characteristic	Variable	Average Score Increase or Decrease	*p*-Value
EORTC QLQ-C30 ^1^	ASA score	Decrease of 18.7–32.4 points	0.04
EORTC QLQ-C30 ^1^	ICU readmission	Decrease of 34.7 points	<0.01
EORTC QLQ-C30 ^1^	Squamous cell carcinoma	Decrease of 7.5 points	<0.01
EORTC QLQ-C30 ^1^	Adjuvant chemotherapy	Decrease of 19.4 points	<0.01
EORTC QLQ-C30 ^1^	Hybrid surgical access	Decrease of 11.4 points	<0.01
EORTC QLQ OES-18	Squamous cell carcinoma	Increase of 6.8 points	0.02
EORTC QLQ OES-18	Neoadjuvant chemotherapy	Increase of 14.1 points	0.02
EORTC QLQ OES-18	Hybrid surgical access	Increase of 8.7 points	0.02
EORTC QLQ OES-18	Open surgical access	Increase of 31.1 points	<0.01
Eckardt (Dysphagia)	ASA score	Increase of 3 points	0.02
GERD-HRQL	Open surgical access	Increase of 12.8 points	0.01
GERD-HRQL	Neoadjuvant chemotherapy	Increase of 8.8 points	0.03
DGCE	Open surgical access	Increase of 4.9 points	0.04
Sigstad (Dumping)	None	-	None

^1^ EORTC QLQ-C30 refers here to the five functional scales and global health (excluding symptom scales). EORTC, European Organization for Research and Treatment; QLQ-C30, Core Quality of Life questionnaire; OES-18, Oesophageal Module; ASA, American Society of Anesthesiologists; ICU, Intensive care unit; GERD-HRQL, Gastroesophageal reflux disease Health-related Quality of Life; DGCE, Delayed Gastric Conduit Emptying.

## Data Availability

The data presented in this study are available on request from the corresponding author. The data are not publicly available due to privacy and ethical restrictions.

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
