# Peer review of "Quality of Life and Independent Factors Associated with Poor Digestive Function after Ivor Lewis Esophagectomy"

_cancers, 2023, doi:10.3390/cancers15235569_

Round 1

Reviewer 1 Report

Comments and Suggestions for Authors

PFA

Reviewer 2 Report

Comments and Suggestions for Authors

This article is a retrospective study on a prospectively gathered database attempting to understand the factors that impact the development of functional digestive complications for patients undergoing Ivor Lewis esophagectomy. While the authors were able to draw significant predictive conclusions for this complicated data set, the comparison seems redundant. To compare post surgery patients to completely healthy individuals in two populations does not validate the complaint or create an adequate comparison. 

1.  There is no mention of IRB or Helsiniki (ethics committee) approval for the research project. 

2. The comparison to two healthy patient populations is not the proper comparison. Would consider comparison of post operative patients to patients with GERD, Gastroparesis and/or dysphagia from other etiologies and demonstrating post operative patient characteristics that may be different. This would help differentiate the post operative complaint from the functional complaint and help with the diagnostic workup. If one would compare to healthy population would perform age, sex and comorbid condition matched controls which would allow for better generalizability. 

3. There seems to be an extensive body of literature on this topic in the past couple of years and would consider a prospective study with at least 1 year follow up to evaluate the course of these functional disorders. 

4. Methods section 2.2 should be moved to the results section 

Reviewer 3 Report

Comments and Suggestions for Authors

The article submitted for review was prepared by practicing doctors who conducted a full-fledged study and summarized their own data on the functional consequences of esophageal cancer surgery. The article extensively discusses individual patient factors, such as surgical approach, comorbidities, and squamous cell carcinoma, that contribute to impairment of quality of life, psychosocial well-being, and physiological function during long-term follow-up. The presented results may be useful for oncologists and surgeons to facilitate therapeutic decision-making and improve treatment pathways, understand which people are at risk for digestive disorders, and advise their patients before surgery for cancer.
